# OCS+:
# IMPROVING PTQ WITH OUTLIER TRANSLATION

## ABSTRACT

Post-training quantization (PTQ) is an effective technique for accelerating DNN model inference, where activations typically follow a bell-shaped distribution. Since commodity hardware employs a linear quantization grid and limited quantization levels, prior PTQs optimize a clipping threshold to minimize overall quantization error, which excludes outliers from the bell-shaped data. However, outliers are non-trivial for low-bit and lightweight models. Thus OCS (Zhao et al., 2019) proposed to save outliers by halving and duplicating. However, in activation quantization, the original OCS sacrifices the precision of the regular inliers, leading to severe accuracy degradation. To address this, we propose OCS+ to save outlier activation without affecting the regular inliers. Consequently, OCS+ theoretically achieves one-bit higher representation under the predefined bitwidth hardware. OCS+ is based on offline mathematical transformation, thus it does not require additional training or re-design works on hardware. Experiments over CNNs and ViTs demonstrate OCS+ significantly outperforms OCS and help improve current PTQ SOTAs, e.g., OCS+ improves the current SOTAs by 12.73% in Acc@1 for W2A2 MobileNet-v2. The code will be released.

## 1 INTRODUCTION

Deep neural networks'(DNN) huge cost has hindered their deployment into real-world applications. To solve this problem, various model compression techniques (Han et al., 2015; Hinton et al., 2015) have been studied. Low-bit model quantization(quant) is one of the commonly used, which generally consists of Quantization-Aware Training (QAT) and Post-Training Quantization (PTQ). PTQ only needs a tiny amount of unlabeled data and does not demand the full training pipeline. Thus PTQ is always the first choice for fast model quantization. Traditional PTQ (Krishnamoorthi, 2018) iteratively searches quant parameters by minimizing the mean squared error(MSE) between FP32 and quantized values. When bitwidth goes lower like 4 bits or 2 bits, these methods suffer from severe accuracy degradation. Recent study proposes to improve low-bit PTQ by quantized-feature reconstruction with gradient descent. AdaRound (Nagel et al., 2020) proposed a layer-by-layer reconstruction, and introduced an adaptive rounding parameters for weight into PTQ reconstruction. BRECQ (Li et al., 2021b) and NWQ (Wang et al., 2022) proposed a block-wise and network-wise reconstruction. QDROP (Wei et al., 2022) proposed to randomly quantize a part of a tensor.

As shown in Fig1, DNN activation usually follows a bell-shaped distribution after training. Due to most commodity hardwares use a linear evenly-spaced quantization levels, we need to decide how to linealy map FP32 values to the limited quantization levels. One naive method is to linearly map the full FP32 range to the full range of quantization levels, which usually causes large quantization error and severe accuracy degradation. One better approach is to make the quantization grids narrower than the FP32 distribution with some criterion like minimizing their Mean Squared Error (MSE)–this is known as clipping. However, clipping reduces overall quantization error at the sacrifices of increasing the distortion on the outliers–it is a trade off that should be carefully optimized. As Tab.1 from SPEQ(Boo et al., 2021), for low-bit and light-weight models, the clipped outlier activations are still helpful if we put them back to model forward inference: the accuracy of one optimized FP32-weight/2bit-Activaion(WFA2) ResNet-20 on CIFAR100 improves as the activation precision increases during inference. Increasing bitwidth during test equals to putting the trained-bitwidth model's outliers back. However, we can not increase bitwidth at will, since most commodity hardware

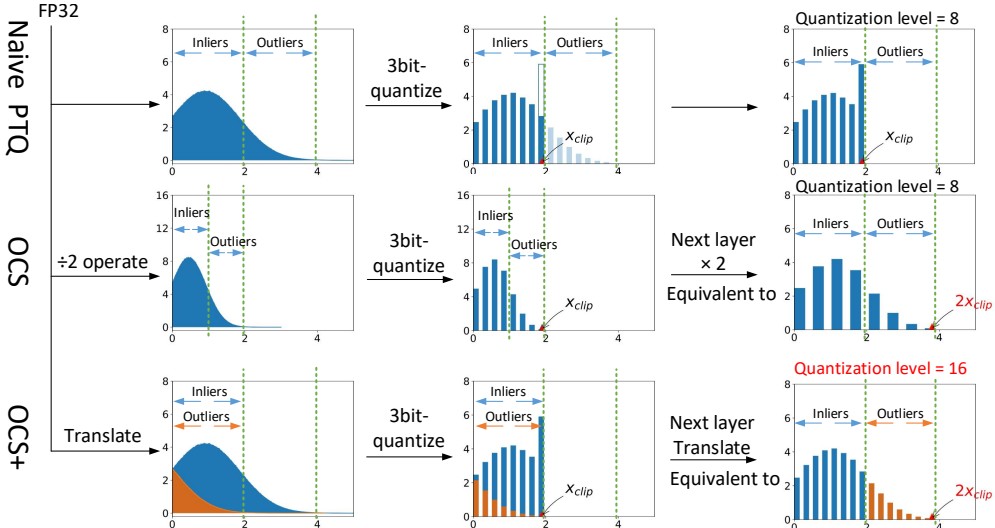

Figure 1: Quantize bell-shaped channel activation into unsigned 3 bits (8 quantization levels). Naive PTQ and OCS both express 8 quantization levels: naive PTQ clips outliers larger than $X_{clip}$; OCS saves outliers in $(X_{clip}, 2X_{clip}]$ but damages regular values in $[0, X_{clip}]$. Our OCS+ achieves 16-quantization- level representation, saving outliers in $(X_{clip}, 2X_{clip}]$ and no damage in $[0, X_{clip}]$.

only supports several predefined bitwidth like 2/4/8/16/32 bits. Therefore, if we can save outliers of an optimized quant-model under predefined bitwidth, the PTQ accuracy can be further improved.

To save outliers, OCS(Zhao et al., 2019) proposed outlier channel splitting by a halving and duplicating operation. As shown in the middle row of Fig1, OCS saves outliers in range $(X_{clip}, 2 \times X_{clip}]$, where $X_{clip}$ is the original optimized quant upper bound, $2 \times X_{clip}$ will be abbreviated as $2X_{clip}$. However, it sacrifices the precision of the regular inliers in range $[0, X_{clip}]$. As a matter of fact, OCS equals to doubling the quant-step of the outlier channels, or making the quantization grid $2\times$ loosely mapped. OCS on activation requires a fixed outlier channel index, but outlier channels of different activation will change as different input, thus OCS on activation will cause accuracy degradation.

To solve this problem, for a well optimized PTQ model, we propose OCS+ to translate, rather than halve, the outliers down into the optimized clipping range and store them in some newly-built channels through a mathematically equivalent transformation on weight as Fig3, so the precision of inliers in range $[0, X_{clip}]$ and outliers in range $(X_{clip}, 2X_{clip}]$ can be both preserved. As the bottom of Fig.1, OCS+ achieves 16-quantization-level representation under 3-bit unsigned quantization, while OCS still owns 8 quantization levels as the naive PTQ. Or to say, OCS save outliers by $2\times$ loosely mapping the quantization grid, while our OCS+ saves outliers with quantization grid still fined-grained mapped. OCS+' saved outliers in range $(X_{clip}, 2X_{clip}]$, or the additional 8 quantization levels, should have been achieved by designing one more bit on hardware. Therefore, OCS+ equals to earning one-more unavailable bit under a predefined-bitwidth hardware. Or to say, we turn the bitwidth increasing re-design work on hardware into an easy (practical) network structure adjustment with some newly-built channels, which only causes tiny inference cost. OCS+ can be applied on common Conv/Linear-$Act$-Conv/Linear structures. $Act$ can be ReLU, H-Swish, GeLU or other nonlinear function. We call them as "OCS-structures". As Tab.8, the fair comparison with the same FLOPs, between OCS+_quantized network and re-trained channels-increased quantized network, demonstrates that OCS+ is not benefit from the new information of newly-added channels, but from the saved outliers in the original channels. Therefore, OCS+ is worthy under a predefined bitwidth hardware after all possible optimization. Our contributions are as follows:

- We propose OCS+, based on offline mathematical transformation, to solve OCS's sacrificing the precision of inliers when saving outliers. With the saved outliers and not-damaged regular inliers, OCS+ makes $b$-bit activation in OCS-structures express $(b + 1)$-bit representations.

- Experiments show OCS+ significantly improves OCS performance and can be easily inserted into existing PTQ libraries to help further improve performance of other PTQ methods.

## 2 RELATED WORK

### 2.1 QUANTIZATION-AWARE TRAINING

Quantization-Aware Training (QAT) requires the whole original training pipeline, including huge amounts of training data, long waiting training hours and complicated quantization knowledge requirements for users. Due to these high consumption, QAT usually performs better than PTQ. Jacob et.al (Jacob et al., 2018) proposed to inject fake quantizers into the original network during the model training, thus the quantization error can be simulated properly and optimized by gradient descent with straight-through estimator (STE). PACT (Choi et al., 2018) proposed parameterized clipping activation to learn the quantization range. One step further, LSQ (Esser et al., 2020) proposed to learn the quantization step directly, which almost achieves fp32's performance in 4 bits. To compensate the gradient mismatch introduced by STE, EWGS (Lee et al., 2021) proposed quantization with element-wise gradient scaling.

### 2.2 POST-TRAINING QUANTIZATION

PTQ takes in a well-trained 32 bit floating-point (FP32) model then covert it into a low-bit fixed-point counterpart directly. For weight quantization, Adaround (Nagel et al., 2020) found that the commonly used rounding-to-nearest operation will be sub-optimal. Thus it proposed an adaptive rounding for weight. BRECQ (Li et al., 2021a) found out block-by-block reconstruction behaves better than the former layer-by-layer ones. NWQ (Wang et al., 2022) further proposed a network-wise PTQ by fully leveraging inter-layer dependency. QDROP (Wei et al., 2022) proposed to jointly optimize quantization error of both weight and activation. Meanwhile, it proposed to randomly quantize only a part of a tensor like dropout. MRECG (Ma et al., 2023) tried to Solve Oscillation problem in PTQ through a theoretical perspective. PD-Quant (Liu et al., 2023) proposed to consider global information based on prediction difference metric. Bit-Shrink (Lin et al., 2023) proposed to limit instantaneous sharpness for improving PTQ. PTQ4ViT (Yuan et al., 2022) and APQ-ViT (Ding et al., 2022) are proposed to solve the PTQ for Vision Transformer(ViT).

The existing clipping operation in PTQ clips the outlier activation beyond the threshold. However, these outliers turns out to be important for low-bit and lightweight models. For this problem, OCS (Zhao et al., 2019) proposed a halving operation to split outlier channels and save outliers without training. As Tab.5, OCS described that it performs well in weight quantization but performs poorly in activation quantization. This is because weight quantization can be done offline, thus we can modify every weight pixel value. so OCS does not cause precision loss for regular weight when preserving outlier weight. However, activation quantization involves real-time computation, thus we only can modify activation per-channel or per-layer, so preserving outlier activations with OCS will damage the precision of regular values, as shown on Fig.1 and Fig.2. Another reason for OCS reporting an 0.1% accuracy on activation quantization is that efficient feature reconstruction had not been introduced. To make fair comparison and follow PTQ feature reconstruction technique with gradient descent, we also re-implement OCS on NWQ as Sec.4.

Differently, we consider feature reconstruction, and propose OCS+ to preserve outlier activation using translation than halving, thus no damage caused on the regular inliers. Such a modified network achieves one-bit higher representation equally in theory on predefined-bitwidth hardware.

## 3 PROPOSED METHODS

Typical linear quantization on activation $x$ is as follows, also known as fake quantization,

$$\hat{\boldsymbol{x}} = clip(\lfloor \frac{\boldsymbol{x}}{s_x} \rceil; x_l, x_u) \cdot s_x \tag{1}$$

where $s_x$ is the quantization step, $x_l, x_u$ are the lower and upper bound of quantization grid, $\lfloor \rceil$ is the rounding operation and $clip$ is to clip the quantized outliers beyond $(x_l, x_u)$ to $x_l$ and $x_u$.

For a predefined hardware, the quantization bitwidth can only be several fixed values, such as 2/4/8. So that $x_l$ and $x_u$ also can only be several fixed values, such as (-2,1)/(-8,7)/(-128, 127). The goal of PTQ is to optimize a suitable quant-step $s_x$ by minimizing the overall quantization error between different de-quantized $\hat{\boldsymbol{x}}$ and FP32 $\boldsymbol{x}$ across the given calibration dataset.

| FP32 | $[x_1 \quad x_2]$ | | $\times \begin{bmatrix} w_1 & w_2 \\ w_3 & w_4 \end{bmatrix} = [y_1 \quad y_2]$ |
|---|---|---|---|
| Naïve 1×Clip | $CLIP\big( \lfloor [x_1 \quad x_2], s_x \rceil,$ | $0, \quad X_{clip} \big)$ | $\times \begin{bmatrix} w_1 & w_2 \\ w_3 & w_4 \end{bmatrix} = [\hat{y}_1 \quad \hat{y}_2]$ |
| OCS | $CLIP\big( \lfloor [x_1 \quad \frac{x_2}{2} \quad \frac{x_2}{2}], s_x \rceil,$ | $0, \quad X_{clip} \big)$ | $\times \begin{bmatrix} w_1 & w_2 \\ w_3 & w_4 \\ w_3 & w_4 \end{bmatrix} = [\hat{y}_1' \quad \hat{y}_2']$ |
| Naïve 2×Clip | $CLIP\big( \lfloor [x_1 \quad x_2], s_x \rceil,$ | $0, \; 2 \times X_{clip} \big)$ | $\times \begin{bmatrix} w_1 & w_2 \\ w_3 & w_4 \end{bmatrix} = [\hat{y}_1'' \quad \hat{y}_2'']$ |
| **OCS+** | $CLIP\big( \lfloor [x_1 \quad x_2 \quad x_2 - X_{clip}], s_x \rceil, 0,$ | $X_{clip} \big)$ | $\times \begin{bmatrix} w_1 & w_2 \\ w_3 & w_4 \\ w_3 & w_4 \end{bmatrix} = [\hat{y}_1'' \quad \hat{y}_2'']$ |

Figure 2: OCS+ V.s. OCS V.s. Clip on 2D outlier channel activation $x_2$ with optimized quant-step $s_x$, upper quant-bound $X_{clip}$. $\lfloor \rceil$ denotes fakely quantizing the channel with $s_x$ as Formula 1. OCS makes the quantization levels coarsely-covered on outlier channels, where the regular inlier values are also affected to be coarsely-covered. OCS+'s quantization level is still fine-grained and equals to "Naive $2 \times X_{Clip}$", better than OCS.

As Tab 1 from SPEQ(Boo et al., 2021), for an optimized Fp32-weight-2bit-Activation (WFA2) quantized ResNet-20 on CIFAR-100, if we increase the activation precision from 2 bits to 4 bits and 6 bits during inference test, the accuracy improves from 66.39% to 68.48% and 68.77%. Increasing bitwidth during test will put the clipped outliers of the trained 2-bit model back to inference. It demonstrates that the outlier activation of an optimized quantized model is helpful to improve accuracy. However, we can not increase bitwidth at will for a given hardware. Therefore, if we can save these outliers without increasing test bitwidth, we can obtain further accuracy improvement.

Table 1: WFA2 quantized ResNet-20 on CIFAR100 test accuracy (%)

| Trained precision | Test accuracy (%) / Inference precision | | |
|---|---|---|---|
| WFA2 | 66.93 / WFA2 | 68.48 / WFA4 | 68.77 / WFA8 |

OCS (Zhao et al., 2019) proposed to save outliers by splitting outlier channels with a halving operation. We borrow OCS' mathematical process on 2D channel activation $x_i$ as Fig.2. Combining Fig.1 and Fig.2, we see OCS saves outliers at the sacrifice of lowering the precision for inliers. Thus OCS' performance is poorer than naive $2 \times X_{Clip}$. For example, given to quantize a outlier channel [0.0, 0.1, 0.2, 0.3, 0.4, 0.5] into unsigned-2 bits with an optimized quant-step 0.1. After quantization and de-quantization, we get as follows: gray color denotes quantization error is caused.

- FP32:              [0.0, 0.1, 0.2, 0.3, 0.4, 0.5];
- Naive Clipping:    [0.0, 0.1, 0.2, 0.3, 0.3, 0.3];
- OCS:               [0.0, 0.0, 0.2, 0.2, 0.4, 0.4];
- **OCS+(ours)**:    [0.0, 0.1, 0.2, 0.3, 0.4, 0.5].

The Rounding error makes OCS lose the regular inliers 0.1 and 0.3: 0.1→ 0.0 and 0.3→0.2. In fact, for activation, OCS equals to make the regular channels remain 1x quant-step, and make the outlier channels 2x quant-step. It achieves two different quant-tep for a per-layer quantized activation. However, activation will change with different input, thus we can not identify the amount and channel-index of the outliers channel in advance. That is why OCS performs poorly on activation.

To solve this problem, we propose OCS+, as process on the bottom of Fig.2, to save outlier activation and preserve the precision of regular inliers, obtaining equal performance as the naive $2 \times X_{Clip}$.

Cascades of $[Conv + Act + Conv]$ is the most common structure in CNNs, as shown in Fig.3(a), which can be denoted as (2, 3). The intermediate output $\boldsymbol{x}^{l+1}$ is quantized into $\hat{\boldsymbol{x}}^{l+1}$ as (4). Here $Act$ can be any activation function like ReLU, GeLU or H-Swish. We take ReLU as example.

$$\text{Conv}^l\text{: } \boldsymbol{y}^l = \boldsymbol{W}^l \boldsymbol{x}^l + \boldsymbol{b}^l, \text{ Act Func:} \boldsymbol{x}^{l+1} = f(\boldsymbol{y}^l), \tag{2}$$

$$\text{Conv}^{l+1}\text{: } \boldsymbol{y}^{l+1} = \boldsymbol{W}^{l+1} \boldsymbol{x}^{l+1} + \boldsymbol{b}^{l+1} \tag{3}$$

$$\hat{\boldsymbol{x}}^{l+1} = clip(\lfloor \frac{f(\boldsymbol{y}^l)}{s_x} \rceil; x_l, x_u) \cdot s_x \to clip(\lfloor \frac{\boldsymbol{y}^l}{s_x} \rceil; 0, x_u) \cdot s_x \tag{4}$$

Owing to the clipping operation, the outlier features, greater than $x_{clip} = s_x \cdot x_u$, in $\boldsymbol{x}^{l+1}$, are clipped to $x_{clip}$, which will cause these outlier part of features unable to be reconstructed or adjusted during PTQ process. Consequently, the final quantized model suffers huge performance drop.

The FP32 upper bound $x_{clip}$ of quantized results:

$$x_{clip} = s_x \cdot x_u = s_x \cdot (2^b - 1) \tag{5}$$

is determined by quant-step $s_x$ and predefined bitwidth $b$. If we want to enlarge $x_{clip}$ and cover more outliers, one way is to enlarge quant-step. However, a larger, or coarser-grained, quant-step will lead to larger discretization error for a converged quantized model. The other way is to enlarge bitwidth, which is impossible for a predefined-bitwidth accelerator. Is there a solution covering more outliers, while requiring the same bit?

Based on the commonly used structure as (a) of Fig.( 3), called as OCS-structures, if we can afford a little more calculation, the FP32 upper bound $x_{clip}$ can be enlarged safely, meanwhile, the quantization bit can be kept as the same. Given that we want to enlarge the FP32 upper bound from $x_{clip}$ to $\beta x_{clip}$, $\beta \geq 1$, Formula (4) is correspondingly transformed to:

$$\hat{\boldsymbol{x}}^{l+1}_{[0, \beta x_{clip}]} = clip(\lfloor \frac{\boldsymbol{y}^l}{s_x} \rceil; 0, x_u) \cdot s_x + ... \tag{6}$$

$$+ clip(\lfloor \frac{\boldsymbol{y}^l - (\beta - 1) \cdot x_{clip}}{s_x} \rceil; 0, x_u) \cdot s_x$$

From the formula above, we can see the outlier activation is translated back to range $[0, X_{clip}]$, which can be achieved through some number (related to $\beta$) of channels added and some simple modification on weight. To simplify denotation, here we set $\beta = 2$. Thus equation 6 can be:

$$\hat{\boldsymbol{x}}^{l+1}_{[0, 2x_{clip}]} = clip(\lfloor \frac{\boldsymbol{y}^l}{s_x} \rceil; 0, x_u) \cdot s_x + \tag{7}$$

$$clip(\lfloor \frac{\boldsymbol{y}^l - x_{clip}}{s_x} \rceil; 0, x_u) \cdot s_x$$

To achieve equation 7, we first need to duplicate activation $\boldsymbol{y}^l$, translate the copied one down by $x_{clip}$ and concatenate them together as follows:

$$\boldsymbol{y}^{l'}_j = \begin{cases} \boldsymbol{y}^l_j & \text{if } 0 \leq j < c_{out} \\ \boldsymbol{y}^l_{j-c_{out}} - x_{clip} & \text{if } c_{out} \leq j < 2c_{out} \end{cases} \tag{8}$$

where $c_{out}$ is the original number of output channels for activation $\boldsymbol{y}^l$. To achieve this operation, we need to transform the original weight $\boldsymbol{W}^l$ and bias $\boldsymbol{b}^l$ to $\boldsymbol{W}^{l'}$ and $\boldsymbol{b}^{l'}$:

$$\boldsymbol{W}^{l'}_{i,j} = \begin{cases} \boldsymbol{W}^l_{i,j} & \text{if } 0 \leq j < c_{out} \\ \boldsymbol{W}^l_{i,j-c_{out}} & \text{if } c_{out} \leq j < 2c_{out} \end{cases}, \tag{9}$$

$$\boldsymbol{b}^{l'}_j = \begin{cases} \boldsymbol{b}^l_j & \text{if } 0 \leq j < c_{out} \\ \boldsymbol{b}^l_{j-c_{out}} - x_{clip} & \text{if } c_{out} \leq j < 2c_{out} \end{cases}$$

With the modified weight $\boldsymbol{W}^{l'}$ and bias $\boldsymbol{b}'$, we can get the results of the Formula equation 8 by $\boldsymbol{y}^{l'} = \boldsymbol{W}^{l'} \boldsymbol{x}^l + \boldsymbol{b}^{l'}$. To ensure the final output is intact as original except for more outliers saved, $\boldsymbol{W}^{l+1}$ needs to be transformed as $\boldsymbol{W}^{l+1'}$:

$$\boldsymbol{W}^{l+1'}_{i,j} = \begin{cases} \boldsymbol{W}^{l+1}_{i,j} & \text{if } 0 \leq i < c_{out} \\ \boldsymbol{W}^{l+1}_{i-c_{out},j} & \text{if } c_{out} \leq i < 2c_{out} \end{cases} \tag{10}$$

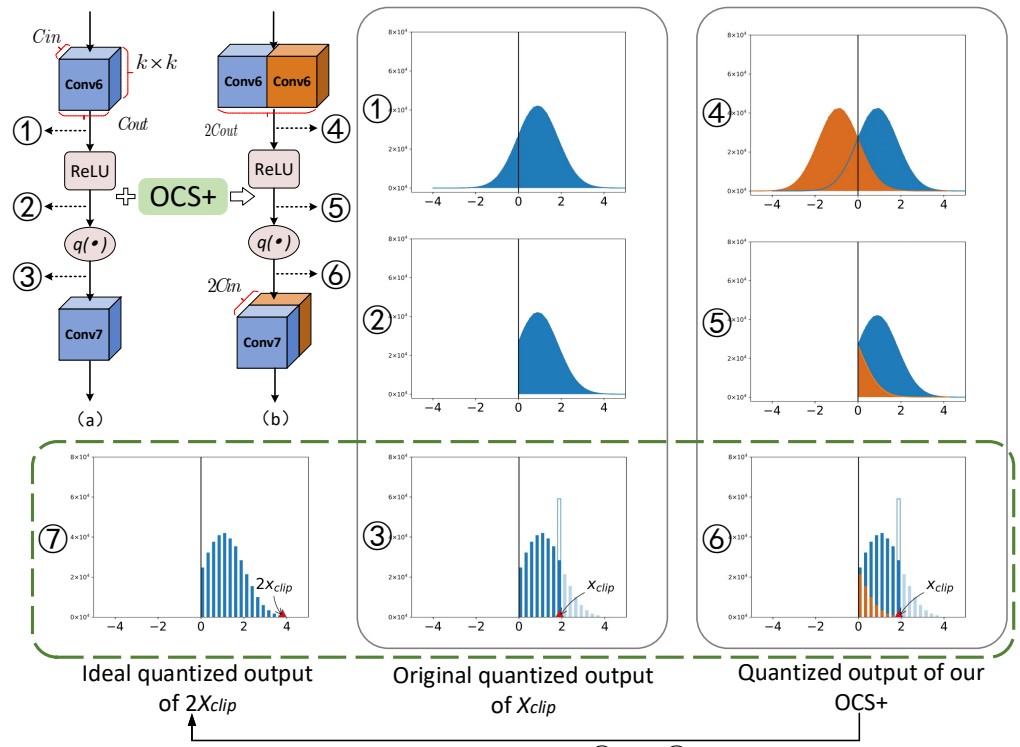

Figure 3: (a) is a typical [*Conv+ReLU+Conv*] structure used in MobileNet-v2. $q(\cdot)$ denotes quantization. Intermediate features of (a) are shown in subgraphs ①②③, whose distributions are shown in the middle. With OCS+ equipped, structure (a) can be transformed to (b), whose intermediate feature distributions are drawn in ④⑤⑥. The orange channels are copied from the blue ones. When the input of Conv7 in a and b is subgraph ⑦ and ⑥ separately, the outputs of Conv7 in a and b are equal. Thus N-bit activation in OCS-structures can be expanded to (N+1)-bit equally.

Such modifications on weights make an functionally identical network except that outlier activation in range $(X_{clip}, 2X_{clip}]$ is preserved. Therefore, our OCS+ enlarges the quantization level and keep the same fine-grained quant-step under the same bitwidth. The OCS process is shown in Algorithm 1.

Besides, not all channels present the same importance. If we can apply our OCS+ merely to the channels whose outliers are the most sensitive to final task loss, a more balanced FLOPs V.s. accuracy trade off can be obtained. Here we assume the proportion of selected channels in each layer as $k \in [0, 1]$. With a given channel sensitivity criterion, all channels can be sorted and the top k percentage of sensitive channels are selected to apply our OCS+. We adopt the sum of the activation in range $[X_{clip}, 2X_{clip}]$ as each channel's sensitivity criterion.

---

**Algorithm 1:** OCS+

---

**Input:** An optimized b-bit Quant-Model$\{W^l\}_{l=1}^N$ with quant-params $s_w, s_x$, activation $x^l$

    # Save outliers on $M$ OCS-structures

**for** $i = 1$ to $M$ OCS-structures **do**

    $X_{clip} = s_x^i * x_u = s_x^i * (2^b - 1)$

    #Modify weight by OCS+ with Formula equation 9

    $W^i \leftarrow concat(W^i, W^i); b^i \leftarrow concat(b^i, b^i - X_{clip})$

    # Modify next layer $W^{i+1}$ as Formula equation 10

    $W^{i+1} \leftarrow concat(W^{i+1}, W^{i+1})$;

**Output:** OCS+ Improved Quantized model

---

# 4 EXPERIMENT

We evaluate OCS+ on ImageNet (Deng et al., 2009) classification and MS COCO (Lin et al., 2014) object detection over various nets and bitwidths using PyTorch (Paszke et al., 2019). The calibration set consists of 1024 (256) unlabeled images randomly selected from the training set. We adopt Adam optimizer, the same learning rate and 20k iterations for network-wise PTQ reconstruction as (Wang et al., 2022). By convention, the first and last layer are quantized into 8 bits.

Experiment on ImageNet(Deng et al., 2009) for CNNs(He et al., 2016) , with average results over 5 runs, are summarized in Tab.2. The proportion of selected important channels in OCS+ is set as $k = 0.5$. Here OCS+ is based on NWQ, denoted as $\mathbf{OCS+}_{0.5\textbf{(ours)}\_\text{NWQ}}$. In W3A3, our method improve Mobile-v2(Sandler et al., 2018) by 2.89%, Reg-600(Radosavovic et al., 2020) by 1.42% and Mnas2.0(Tan et al., 2019) by 2.87%. In W2A2, BRECQ shows nearly 0% on Mobile-v2 and Mnas2.0, but our OCS+ still far outperforms current PTQs, e.g., 12% better than NWQ on Mobile-v2.

Table 2: Acc@1 on ImageNet among current PTQ methods.

| Methods | W/A | MobileNet-v2 | ResNet-18 | RegNet-600 | MnasNet2.0 |
|---|---|---|---|---|---|
| Full Prec. | 32/32 | 72.49 | 71.08 | 73.71 | 76.68 |
| BRECQ(Li et al., 2021b) | 4/4 | 66.57 | 69.60 | 68.33 | 73.56 |
| QDROP(Wei et al., 2022) | 4/4 | 68.84 | 69.62 | 71.18 | 73.71 |
| PD-Quant (Liu et al., 2023) | 4/4 | 68.33 | 69.30 | 71.04 | 73.30 |
| MRECG (Ma et al., 2023) | 4/4 | 68.84 | 69.46 | 71.22 | - |
| NWQ (Wang et al., 2022) | 4/4 | 69.14 | 69.85 | 71.92 | 74.60 |
| $\mathbf{OCS+}_{0.5\_\text{NWQ(ours)}}$ | 4/4 | $\mathbf{70.19}_{\pm0.14}$ | $\mathbf{70.10}_{\pm0.05}$ | $\mathbf{72.34}_{\pm0.12}$ | $\mathbf{75.14}_{\pm0.24}$ |
| BRECQ(Li et al., 2021b) | 3/3 | 23.41 | 65.87 | 55.16 | 49.78 |
| QDROP(Wei et al., 2022) | 3/3 | 57.98 | 66.75 | 65.54 | 66.81 |
| PD-Quant (Liu et al., 2023) | 3/3 | 57.64 | 66.12 | 65.09 | 64.88 |
| MRECG (Ma et al., 2023) | 3/3 | 58.40 | 66.30 | 66.08 | - |
| NWQ (Wang et al., 2022) | 3/3 | 61.24 | 67.58 | 67.38 | 68.85 |
| $\mathbf{OCS+}_{0.5\_\text{NWQ(ours)}}$ | 3/3 | $\mathbf{64.13}_{\pm0.16}$ | $\mathbf{68.20}_{\pm0.06}$ | $\mathbf{68.80}_{\pm0.11}$ | $\mathbf{71.72}_{\pm0.22}$ |
| BRECQ(Li et al., 2021b) | 2/2 | 0.24 | 42.54 | 3.58 | 0.61 |
| QDROP(Wei et al., 2022) | 2/2 | 13.05 | 54.72 | 41.47 | 28.77 |
| PD-Quant (Liu et al., 2023) | 2/2 | 13.67 | 53.14 | 40.92 | 28.03 |
| MRECG (Ma et al., 2023) | 2/2 | 14.44 | 54.46 | 43.67 | - |
| NWQ (Wang et al., 2022) | 2/2 | 26.42 | 59.14 | 48.49 | 41.17 |
| $\mathbf{OCS+}_{0.5\_\text{NWQ(ours)}}$ | 2/2 | $\mathbf{39.15}_{\pm1.68}$ | $\mathbf{61.46}_{\pm0.23}$ | $\mathbf{54.73}_{\pm0.27}$ | $\mathbf{50.83}_{\pm1.25}$ |

ViT (Dosovitskiy et al., 2021) and DeiT (Touvron et al., 2021) experiments is as Tab.3. Here OCS+ is based on OAS, denoted as $\mathbf{OCS+}_{0.5\textbf{(ours)}\_\text{OAS}}$ Note that OCS+ support ViT and DeiT's GeLU activation function. OCS+ further improves OAS(Ma et al., 2024)'s performance by 2.03% on ViT-S, 1.86% on ViT-B, 1.21% on DeiT-S and 1.01% on DeiT-B.

Table 3: Acc@1 on ImageNet for ViTs and DeiTs.

| Methods | W/A | ViT-S | ViT-B | DeiT-S | DeiT-B |
|---|---|---|---|---|---|
| FP32 | 32/32 | 81.39 | 84.54 | 79.80 | 81.80 |
| PTQ4ViT (Yuan et al., 2022) | 4/4 | 42.57 | 30.69 | 34.08 | 64.39 |
| APQ-ViT (Ding et al., 2022) | 4/4 | 47.95 | 41.41 | 43.55 | 67.48 |
| NWQ (Wang et al., 2022) | 4/4 | 57.79 | 56.87 | 65.76 | 76.06 |
| RepQ-ViT (Li et al., 2023) | 4/4 | 65.05 | 68.48 | 69.03 | 75.61 |
| ERQ (Zhong et al., 2024) | 4/4 | 68.91 | 76.63 | 72.56 | 78.23 |
| OAS (Ma et al., 2024) | 4/4 | 72.88 | 76.59 | 76.00 | 78.83 |
| $\mathbf{OCS+}_{0.5\_\text{OAS(ours)}}$ | 4/4 | $\mathbf{74.91}$ | $\mathbf{78.45}$ | $\mathbf{77.21}$ | $\mathbf{79.84}$ |

Object detection results are as Tab.4. Here OCS+ is based on NWQ, denoted as $\textbf{OCS+}_{0.5\textbf{(ours)}\_\text{NWQ}}$. As (Wei et al., 2022; Li et al., 2021b), we quantize the input and output layers to 8 bits, do not quantize detection head, and quantize neck (FPN). In W3A3 setting, OCS+ improves Res-50-based Faster RCNN by 1.23% and Mobile-v2-based RetinaNet by 1.30%. In harder W2A2 setting, OCS+ improves more than 3% mAP over the current best method across all four networks.

Table 4: mAP on MS COCO for object detection.

| Methods | W/A | Faster RCNN(Ren et al., 2015) | | RetinaNet(Lin et al., 2017) | |
|---------|-----|-------------------------------|--|-----------------------------|--|
| | | ResNet-50 | ResNet-18 | ResNet-50 | MobileNet-v2 |
| FP32 | 32/32 | 40.26 | 34.91 | 37.39 | 33.31 |
| QDROP (Wei et al., 2022) | 4/4 | 38.53 | 33.57 | 35.81 | 31.47 |
| NWQ (Wang et al., 2022) | 4/4 | 38.54 | 33.63 | 35.98 | 31.81 |
| **OCS+$_{\textbf{0.5}\_\text{NWQ(ours)}}$** | 4/4 | **38.94** | **34.08** | **36.10** | **32.16** |
| QDROP (Wei et al., 2022) | 3/3 | 33.49 | 31.21 | 32.13 | 27.55 |
| NWQ (Wang et al., 2022) | 3/3 | 35.25 | 31.88 | 32.45 | 28.43 |
| **OCS+$_{\textbf{0.5}\_\text{NWQ(ours)}}$** | 3/3 | **36.48** | **32.56** | **33.24** | **29.73** |
| QDROP (Wei et al., 2022) | 2/2 | 21.05 | 21.95 | 20.27 | 12.01 |
| NWQ (Wang et al., 2022) | 2/2 | 25.01 | 23.92 | 22.95 | 16.21 |
| **OCS+$_{\textbf{0.5}\_\text{NWQ(ours)}}$** | 2/2 | **29.82** | **27.30** | **26.20** | **20.25** |

## 5 ABLATION STUDY

### 5.1 OCS+ V.s. OCS:

As OCS(Zhao et al., 2019) described, it did a good job in weight quantization but failed to quantize activation into low bits, denoted as $OCS_{\text{ori}\_0.05}$ as Tab 5, where 0.05 denotes selected outlier channels ratio $k = 0.05$. To make fair comparison and follow current efficient PTQ reconstruction, we re-implement OCS on activation with NWQ in ResNet-50, denoted as $OCS_{\text{new}}$ and compare with our OCS+ as Tab 5. We can see NWQ significantly improves OCS performance on W8A4 from 0.1% to 75.26% when selected outlier channel ratio $k = 0.05$. On W4A4, $k = 0.05$, OCS+ is 0.16% better than $OCS_{\text{new}}$. When bitwidth goes lower, the OCS+'s benefit than OCS is obvious. As Tab 6, on W2A2, in ResNet-50, MobileNet-V2 and MnasNet-2.0, a) $k = 0.1$, OCS+ is better than $OCS_{\text{new}}$ by 1.4%, 1.6%,2.5%. b) $k = 0.5$, OCS+ is better than $OCS_{\text{new}}$ by 8.0%, 13.4%,13.0%. We can see $OCS_{\text{new}}$ is worse than the baseline NWQ and OCS's performance decreases as the selected outlier channel ratio increase. The reason is that, for a per-layer quantized activation (one single quant-step), OCS achieves 1x quant-step for regular channels and 2x quant-step for outlier channels, thus if we carefully choose the outlier channels whose quant-error is lower in 2x quant-step than in 1x quant-step, we will get better performance. However, if we choose more outlier channels over a threshold, the optimization objective will be shifted away from optimal point and cause worse performance than baseline. Differently, our OCS+ will not collapse the quantization performance.

Table 5: Original/New OCS V.s. OCS+

| Methods | W/A | Acc@1 |
|---------|-----|-------|
| $OCS_{\text{ori}\_0.05}$(Zhao et al., 2019) | 4/8 | 69.3 |
| $OCS_{\text{ori}\_0.05}$(Zhao et al., 2019) | 8/4 | 0.1 |
| $OCS_{\text{new}\_0.05}$ | 8/4 | 75.26 |
| $OCS_{\text{new}\_0.05}$ | 4/4 | 75.22 |
| **OCS+$_{0.05\textbf{(ours)}}$** | 4/4 | **75.38** |

Table 6: New OCS V.s. OCS+ on W2A2 Nets

| Methods | Res-50 | Mobile-V2 | Mnas2.0 |
|---------|--------|-----------|---------|
| NWQ | 57.18 | 26.42 | 41.17 |
| $OCS_{\text{new}\_0.1}$ | 56.31 | 26.11 | 41.12 |
| **OCS+$_{0.1\textbf{(ours)}}$** | **57.76** | **27.74** | **43.65** |
| $OCS_{\text{new}\_0.5}$ | 52.82 | 25.74 | 37.87 |
| **OCS+$_{0.5\textbf{(ours)}}$** | **60.83** | **39.15** | **50.83** |

Their practical extra-introduced cost is the same. Tab.7 shows the fair inference comparison with expanding ratio 0.2. Compared to baseline NWQ, OCS+ improve 1.0% accuracy with only additional 0.01 ms. Compared to NWQ-applied-OCS, OCS+ improves 2.2% accuracy with the same 0.17 ms.

Table 7: MobileNet-V2 W4A4 Inference Cost

| Method | Acc@1 | Infer-Time / Per sample | Params | FLOPs*Bit |
|---|---|---|---|---|
| Baseline - NWQ | 69.14 | 0.16 ms | 3.51 M | 6.36 G |
| $OCS_{0.2}$ - NWQ | 68.90 | 0.17 ms | 3.75 M | 6.95 G |
| **OCS+$_{0.2(ours)}$ - NWQ** | **70.10** | 0.17 ms | 3.75 M | 6.95 G |

## 5.2 WHETHER OCS+ GAINS FROM EXTRA-CHANNELS OR FROM SAVED-OUTLIERS?

For OCS+$_{0.5}$, it brings extra 50% channels on OCS-structures. Thus we add the same number of channels on the same structures in FP32 networks to test whether accuracy gain comes from extra channels. We train these new FP32 networks from scratch with timm (Wightman, 2019) training pipeline. As the second row of Tab.8, extra channels bring extra Acc@1 gain for FP32 networks. Then we quantize the finetuned new FP32 Nets and the original FP32 Nets with NWQ, but the latter is additionally modified by OCS+. As Tab.8, OCS+$_{0.5}$ achieves better performance across networks and bitwidths, especially for Mobile-v2, about 5.5% better in W4A2 and 7.1% better in W2A2. **So, the gain, from obtaining better FP32 accuracy by training a new more-channel-added FP32 network, will get lost after using current PTQ methods.** Therefore, our OCS+ does not gain from extra channels, but from extra saved outliers.

Table 8: Acc@1 of Channel-plus-retrained Net and OCS+ saved-outliers Net on ImageNet

| Methods | W/A | MobileNet-v2 | ResNet-18 | RegNet-600 | MnasNet-2.0 |
|---|---|---|---|---|---|
| Ori-Net | 32/32 | 72.49 | 71.08 | 73.71 | 76.68 |
| Channels-Increase-Net | 32/32 | 74.85 | 71.80 | 75.37 | 78.57 |
| **OCS+$_{0.5}$_Ori-Net** | 4/2 | **48.78** | **63.72** | **62.86** | **58.56** |
| NWQ_Channels-Increase-Net | 4/2 | 43.24 | 62.09 | 60.20 | 53.23 |
| **OCS+$_{0.5}$_Ori-Net** | 2/2 | **39.15** | **61.46** | **54.73** | **50.80** |
| NWQ_Channels-Increase-Net | 2/2 | 32.02 | 60.64 | 52.73 | 47.24 |

## 5.3 EXPLORE DIFFERENT PROPORTION $k$ IN OCS+

W2A2 experiments on different networks as Tab.9 with expanding ratio $k$ in [0,0.3,0.5,0.7,1.0] shows accuracy improves with OCS+ applied, and accuracy gain improves as $k$ increases.

Table 9: Acc@1 among different $k$ of OCS+

| Methods | W/A | MobileNet-v2 | ResNet-18 | RegNet-600 | MnasNet2.0 |
|---|---|---|---|---|---|
| **OCS+$_{0.0}$(ours)** | 2/2 | 26.42 | 59.14 | 48.49 | 41.17 |
| **OCS+$_{0.3}$(ours)** | 2/2 | 35.79 | 61.23 | 53.17 | 48.20 |
| **OCS+$_{0.5}$(ours)** | 2/2 | 39.15 | 61.46 | 54.73 | 50.83 |
| **OCS+$_{0.7}$(ours)** | 2/2 | 39.77 | 61.88 | 55.86 | 53.32 |
| **OCS+$_{1.0}$(ours)** | 2/2 | **41.49** | **62.20** | **57.31** | **54.38** |

## 6 CONCLUSION

Considering the failure of OCS on activation quantization and the development of PTQ feature reconstruction, in this paper, we propose OCS+, through translation instead of halving, to preserve outlier activation without affecting regular inlier one. Thus $b$-bit activation in OCS-structures can be theoretically expanded to $(b + 1)$ bits. With the same additional computation as OCS, OCS+ significantly outperforms OCS, especially on 2/3bit and light-weight models. OCS+ can be easily inserted into existing PTQ libraries to help further improve performance on activation quantization.

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
