# OpenReview forum: "OCS+: Improving PTQ with Outlier Translation"
_ICLR.cc/2025/Conference — Submitted to ICLR 2025_

### Official Review · Reviewer_QVvG · 2024-10-29

**Soundness:** 3
**Presentation:** 3
**Contribution:** 1
**Rating:** 3
**Confidence:** 5

**Summary:**

This paper introduced OCS+, a PTQ for solving the outlier in the quantization process. First, this paper demonstrated that outliers are non-trivial. Motivated by OCS, the original version of this paper, OCS+ is introduced. OCS+ duplicates the important (outlier) activation channel and corresponding weight. Thus, OCS+ achieves high performance with the costs of more computation.

**Strengths:**

1. This paper provided comprehensive experimental results.

2. This paper has clear logic that makes their method easy to understand.

**Weaknesses:**

1. Lack of necessary fair comparison. Actually, OCS+ introduces higher computation costs since it increases the weight and activations by 50%. It added 50% extra computation. This makes OCS+ less attractive since the compared methods do not incur extra computation. Also, some related descriptions such as the analysis of FLOP (compared with previous methods)are lacking in the paper.

2. How to select the important activation channels missed in the paper?

3. The implementation details of Table 7 are missing. What are the platform and running software?

4. Running time comparison with previous methods that do not incur extra computation such as BRECQ?

5. According to the paper, the important activation channels seem to be selected on the fly. Does this operation incur more inference time costs?

**Questions:**

I am wondering what the point of extremely low-bit quantization is, such as 2-bit. Does this extremely low-bit make any practical use? Could the author provide some insight into this?

I currently turn to reject this paper for its novelty and experiment comparison.

---

### Official Review · Reviewer_Rcu4 · 2024-11-01

**Soundness:** 3
**Presentation:** 2
**Contribution:** 3
**Rating:** 5
**Confidence:** 5

**Summary:**

This paper proposes a post-training quantization method named OCS+, which aimed at saving outlier activation without affecting the regular inliers. Based on the offline transformation with weight and activation, it does not require additional training. Experimental results show that the proposed method works for both CNN and ViTs.

**Strengths:**

1.The proposed OCS+ preserves the outlier activations without sacrificing the precision of regular inliers, which allows for a theoretical increase in presentational power from b-bits to (b+1)-bits under the same hardware constraints.

2.OCS+ is based on the offline mathematical transformations, which does not require additional training or hardware re-design.

3.Experimental results show OCS+ achieves an improvement over the previous methods both on CNN and ViTs.

**Weaknesses:**

1.The presentation need to be improved, such as Page 4.

2.The actual speed-up need to be evaluated since  OCS+ introduces additional computational costs.

3.Some typos: such as
Line 93 and Line 99 OCS+-.

Line 36 quant parameters

Line 253 translate down?

**Questions:**

See weaknesses.

---

### Official Review · Reviewer_jQCD · 2024-11-03

**Soundness:** 3
**Presentation:** 3
**Contribution:** 2
**Rating:** 5
**Confidence:** 4

**Summary:**

Outliers while quantization are one of the representative elements that harm the performance of neural networks.
What makes things worse is that target hardware is usually fixed and hard to change settings such as bit precision.
Thus, if the outlier problem becomes severe and the performance of the quantized model deteriorates, it may be difficult to resolve the problem by using a higher bit.

There already exists a prior work that tries to resolve this problem which is called OCS.
By halving the activation values of outlier channels and duplicating those channels as additional channels, OCS successfully alleviates the outlier problem.
However, OCS causes another problem; rounding error of inliers.

To mitigate both problems (clipping error of outliers and rounding error of inliers) simultaneously, the paper proposes OCS+.
The paper adopts translation instead of halving operation.
By doing so, it achieves the same functional effect equivalent to using one more bit with moderate computational overhead.
With various experiments, the paper shows that OCS+ outperforms other previous works, even with the same number of channels added by OCS+.

**Strengths:**

- The paper highlights the problem of previous work, OCS.
- The paper shows performance gain compared to OCS.
- Experimental design and multifaceted analysis of the proposal are commendable.
- Under the situations that the paper assumes (e.g., the target hardware and bit settings are fixed), significant performance improvements are expected with additional computation overhead.

**Weaknesses:**

- Several problems that OCS already has can be the same problems of OCS+.
  - The computational overhead due to additional channels
  - The purpose of quantization is to run a large model on limited resources. Therefore, additional computational overhead induced by OCS+ has a worse impact on hardware that is hard to adjust bit precision, which is the target of OCS+.
- The proportion of channels suffering the outlier problem and outlier channel ID can differ according to inputs. Analysis of channel sensitivity with different inputs can be a good experiment.

**Questions:**

In Table 1, are those 3 results attained from the same weight parameters and clipping range? Otherwise, are the clipping ranges of the results different?

---

### Meta-Review · Area_Chair_UZgm · 2024-12-20

**Metareview:**

Although this work aims at addressing the outlier activation preservation issue, the drawbacks related to computational cost, limited novelty, and lack of comprehensive comparisons and essential details outweigh its contribution. All reviewers gave negative scores and there is no response to their concerns. Therefore, this work is recommended to be rejected.

**Additional Comments On Reviewer Discussion:**

There were no discussions between reviewers and authors because authors did not respond to comments from reviewers.

---

### Decision · Program_Chairs · 2025-01-22

Reject